# Plants, Plants, and More Plants: Plant-Derived Nutrients and Their Protective Roles in Cognitive Function, Alzheimer’s Disease, and Other Dementias

**DOI:** 10.3390/medicina58081025

**Published:** 2022-07-30

**Authors:** Helen Ding, Allison B. Reiss, Aaron Pinkhasov, Lora J. Kasselman

**Affiliations:** NYU Long Island School of Medicine, Mineola, NY 11501, USA; helen.ding@nyulangone.org (H.D.); allison.reiss@nyulangone.org (A.B.R.); aron.pinkhasov@nyulangone.org (A.P.)

**Keywords:** Alzheimer’s disease, plant-based diet, B vitamins, antioxidants, vitamin K, fiber, cognition

## Abstract

*Background and Objectives:* Alzheimer’s disease (AD) is the most common form of dementia, with the risk of developing it attributed to non-modifiable and modifiable factors. Currently, there is no cure for AD. A plant-based diet may protect against cognitive decline, due to the effects of plant-based nutrients such as vitamins, antioxidants, and fiber. The aim of the review is to summarize current literature on plant-based nutrients and their impact on cognition. *Materials and Methods:* A search was conducted on PubMed for clinical and murine studies, using combinations of the following words: “Alzheimer’s disease”, “dementia”, “cognition”, “plant-based diet”, “mild cognitive impairment”, “vitamin B”, “vitamin C”, “vitamin E, “beta carotene”, “antioxidants”, “fiber”, “vitamin K”, “Mediterranean diet”, “vitamin D”, and “mushrooms”. *Results and Conclusions:* A diet rich in vitamin B and antioxidants can benefit the cognitive functions of individuals as shown in randomized clinical trials. Vitamin K is associated with improved cognition, although large randomized controlled trials need to be done. Fiber has been shown to prevent cognitive decline in animal studies. Vitamin D may contribute to cognitive health via anti-inflammatory processes. Several medical organizations have recommended a plant-based diet for optimizing cognitive health and potentially helping to prevent dementia.

## 1. Introduction

Alzheimer’s disease (AD) is the most common form of dementia and accounts for 60–80% of all dementia cases [1]. AD impacts more than 40 million people worldwide, and the prevalence of AD doubles every five years after age 65 [2,3]. While there is a normal age-related decline in memory that does not impact activities of daily living, an accelerated decline that interferes with a person’s quality of living is seen in Alzheimer’s disease [4]. Alzheimer’s disease is often preceded by a period of mild cognitive impairment (MCI) [5]. MCI is defined as cognition that is no longer normal relative to age expectations but does not interfere with activities of daily living, and it is reported that MCI affects more than 40% of the population over age 60 [6]. AD poses a significant burden to patients, namely the number of healthy years lost due to being in a state of disability as well as premature mortality [7]. AD also poses a burden to caregivers, with increased caregiver burden associated with severity of AD [8].

The onset of AD is predicted by multiple risk factors. Some non-modifiable factors include age, gender, and genetic susceptibility such as apolipoprotein E status [9]. The cause of AD is likely multifactorial and the current understanding of the pathophysiology underlying MCI and AD is still incomplete. The most widely studied theory of AD pathogenesis is the accumulation of neurotoxic extracellular beta-amyloid plaques, which are insoluble protein aggregates that are pathological hallmarks of AD [10]. While the pathogenesis of AD is still not fully understood, the main hypothesis is that beta-amyloid plaques interact with microglia in a way that results in pro-inflammatory cytokines and reactive oxygen species, which contribute to neuronal damage [11]. Furthermore, continuously elevated levels of beta-amyloid cause activation of the innate immune system via microglia activation, which leads to an inflammatory cascade that may contribute to AD pathogenesis [12].

The failure of anti-amyloid therapies to change the course of AD in human trials has caused a shift in thinking about the etiopathogenesis of the disease [13,14,15]. Inflammation and oxidative stress may result in synaptic and neuronal loss via mechanisms independent of amyloid formation [16,17]. High levels of inflammatory markers are often present in AD patients, and these same markers are associated with cognitive decline [18]. In addition to increased inflammation, which may be toxic long-term, AD has been associated with mitochondrial dysfunction, with changes in mitochondrial processing resulting in increased oxidative stress [19]. Neurons are highly dependent on mitochondria, as they have a high metabolic demand. High levels of reactive oxygen species (ROS) can build-up as a result of mitochondrial dysfunction or an insufficient amount of antioxidants, which leads to oxidative stress [20]. Furthermore, increased ROS levels result in increased inflammatory markers, demonstrating the connection between oxidative stress and inflammation [21].

In addition to beta-amyloid proteins, the microtubule-associated protein tau has been heavily studied in its relation to Alzheimer’s disease [22,23]. Tau proteins in AD are hyperphosphorylated, causing the proteins to stick together in neurofibrillary tangles inside neurons [24]. These tangles then interfere with chemical and electrical signaling between neurons, and disruption of this process can lead to dysfunctional synapses and neuronal death [25]. Mouse models of AD have also linked tau to beta-amyloid build-up, hypothesizing that abnormalities of tau phosphorylation contribute to the toxicity of beta-amyloid [26]. Beta-amyloid’s effects on neurotransmitter levels is well-documented in current literature, with studies showing beta-amyloid enhancing glutamate uptake and inhibiting acetylcholine release [27,28,29].

Current pharmacological interventions for Alzheimer’s disease aim to correct the neurotransmitter imbalances that likely result from tau protein build up and neuronal dysfunction [30]. Current FDA approved drugs to treat cognitive symptoms of AD are acetylcholinesterase inhibitors (AChEI) and a glutamate regulator, memantine. Acetylcholinesterase inhibitors increase levels of acetylcholine, a neurotransmitter important in memory, attention, and learning [30]. Memantine decreases levels of glutamate, which is thought to be involved in neurotoxicity seen in AD [4,31]. These two classes of drugs are only effective in treating AD symptoms, but do not prevent or cure it. Furthermore, AChEI drugs have shown low efficacy in improving cognition of patients with AD, and they are associated with multiple adverse effects, like diarrhea, nausea, vomiting, bradycardia, and syncope [32]. Given the arguable efficacy of AChEIs and their side effects, the risk-benefit relationship of the drug is unclear. Studies on memantine have unclear results, with some clinical trials demonstrating meager evidence of its treatment in AD [33,34]. The complexity of AD etiopathology makes it difficult to prevent and cure the disease. Furthermore, current medications attack downstream phenomena, like neurotransmitter imbalances, that do not directly address the build-up of beta-amyloid and tau proteins, oxidative stress, or inflammation that are hypothesized to drive AD progression [4,18,21,35]

Although there is no cure for Alzheimer’s disease, individuals can reduce their risk for developing AD by targeting modifiable risk factors. There are several modifiable risk factors for AD, such as diabetes, obesity, smoking, and hypertension. Additionally, modifiable risk factors like physical activity and healthy diet decrease the risk of developing AD [36]. Recent research has supported a diet rich in fruits and vegetables to be associated with prevention and delay of cognitive impairment [37]. Plant-based diets have been shown to be rich in antioxidants, vitamins, and fiber. Antioxidants and vitamins protect against neuronal degeneration via their anti-inflammatory effects and prevention of oxidative stress. Specific plant-derived nutrients are associated with decreased MCI, namely vitamins B, K, C, E and beta-carotene, the latter three being antioxidants [38,39]. The current hypothesis on fiber’s role in cognitive impairment focuses on the gut–brain axis, with fiber promoting certain bacteria in the gut microbiome that influence brain health and neuroinflammation [40].

In this review paper, we will examine the evidence of plant-derived vitamins B, K, C, E, beta-carotene, and fiber and their roles in preventing or delaying cognitive decline, as seen in dementias like Alzheimer’s disease. Though technically not a plant, mushrooms will be briefly reviewed with respect to vitamin D and its role in cognitive health. This review is not exhaustive of all the benefits that these nutrients offer. Given the high prevalence of MCI in the elderly population and the safety and cost effectiveness of diet-change, we hope that this paper will be an extension of the current evidence supporting plant-based nutrition as a tool to reduce the risk of Alzheimer’s disease [41,42,43].

## 2. Materials and Methods

A search was conducted on PubMed for epidemiological, clinical, and animal studies using the following keywords with the Booleans “AND” and “OR” in different combinations: “Alzheimer’s disease”, “dementia”, “cognition”, “plant-based diet”, “mild cognitive impairment”, “vitamin B”, “vitamin C”, “vitamin E, “beta carotene”, “antioxidants”, “fiber”, “vitamin K”, “Mediterranean diet”, “vitamin D”, and “mushrooms”. Only human studies were reviewed in the discussions of vitamin B, antioxidants, and vitamin K interventions, although murine models were used in the discussion of mechanism. Both human and murine models were reviewed in the discussion of fiber. Publications in languages other than English were excluded. This review prioritized larger and more recent publications. 

## 3. Results

### 3.1. Vitamin B

The B vitamins are made up of eight water-soluble vitamins (B1, B2, B3, B5, B6, B7, B9, B12) that act as coenzymes in many catabolic and anabolic reactions. The B vitamins’ role in the brain includes synthesis of neurochemicals and production of methyl groups, which are necessary for DNA/RNA formation and repair [44]. Human epidemiologic studies have focused on the cognitive health benefits of vitamin B in the context of homocysteine metabolism [45]. Homocysteine is a risk factor for both cardiovascular disease and brain atrophy, and plasma homocysteine levels have been shown to be lowered with administration of vitamin B6, folate (vitamin B9), and vitamin B12 [45,46,47]. Vitamin B12 plays a role in the transformation of homocysteine to the amino acid methionine, and B6 and folic acid are necessary cofactors in that reaction [48].

#### 3.1.1. Dietary Sources of Vitamin B 

Most of the current literature on vitamin B and cognitive function focuses on vitamins B6, folic acid, and B12. As such, this review will focus specifically on these three subtypes, of which B6 is most bioavailable from plants. High levels of these B vitamin types have been related to higher cognitive performance, due to their homocysteine lowering effects [49]. Each of these types of B vitamins have different dietary sources, both plant and non-plant based.

The usual dietary sources of vitamin B12 are from animal products, such as meat, milk, fish, and eggs [50,51]. Vitamin B12 is synthesized by certain bacteria and archaea that are present in the gut of animals, but not in plants. After animals like cattle acquire B12 in their gut, the vitamin accumulates in tissue, which makes meat one of the best sources of B12 [52]. While vitamin B12 has been proposed to play an important role in Alzheimer’s disease prevention, it is not plant-derived and as such would not be increased in a plant-based diet [53]. However, because current studies examine B12, B9, and B6 together, we still are discussing the role of B12 on cognitive function.

Folate is the natural form of vitamin B9, and folic acid is the synthetic form of folate that is found in fortified foods, like rice, pasta, and cereals. Natural folate is found in plant-based foods, particularly tropical fruits like mango and kiwi and green leafy vegetables, however folate has a lower bioavailability than synthetic folic acid [54,55]. The limited bioavailability of folate is due to luminal factors like its destruction in the gastrointestinal tract and its absorption variability [56]. On the other hand, these factors do not impact the absorption of synthetic folic acid. Folic acid fortified foods have been shown to be up to two times more bioavailable than naturally occurring folate [56]. The various factors that impact naturally occurring folate absorption makes its bioavailability variable, and much of the dietary folate is from fortified non-plant foods [57].

Vitamin B6 is widely present in many foods, including meat, fish, beans, grains, fruits, and vegetables [58]. Its absorption in the intestine is via passive diffusion, which makes it rapidly absorbable [59]. A major source of vitamin B6 is through plants, specifically in chickpeas, potatoes, bananas, and squash [60]. Due to B6′s high bioavailability from plants, it is the B vitamin subtype that would be most implicated in a plant-based diet. Although B12 is solely derived from animal products, B9 can be found in some plant-based foods, such as beans and avocado. These foods also contain B6, although B6 is more rapidly absorbable than B9. 

#### 3.1.2. Vitamin B Interventions and Current Research

Of all the B vitamins, B6 is the most bioavailable subtype from plant foods. However, there is currently a lack of research focusing solely on B6′s role in cognition. Current research lumps B12, B9, and B6 together as they are cofactors in the metabolism of homocysteine, which has neurotoxic effects. As such, the studies in this section will not delineate between the various B vitamin subtypes.

Vitamin B6 is an important co-factor in the breakdown of homocysteine [61]. Elevated levels of homocysteine are a strong modifiable risk factor for vascular dementia and Alzheimer’s disease [62,63]. High homocysteine levels are also associated with cognitive decline, brain atrophy, and neurofibrillary tangles [62]. A 2016 clinical trial by Cheng et al. found that supplementation of vitamin B improved cognitive function in patients with hyperhomocysteinemia [64]. This clinical trial found that daily vitamin B supplementation of 800 µg of folate, 10 mg of B6 and 25 µg of B12 resulted in improved cognitive function and reduced homocysteine levels after 14 weeks. Similarly, de Jager et al. found that patients with MCI had significant improvements in global cognition, episodic memory, semantic memory, and reduction in total homocysteine, with vitamin B treatment [65]. 

Another randomized control trial found no effect of vitamin B supplementation (2.5 mg folic acid, 0.4 mg B12, 25 mg of B6) on beta amyloid protein levels [66]. Total serum homocysteine correlated with plasma beta amyloid levels, and while participants in this study had significantly decreased homocysteine, there was interestingly no change in plasma levels of beta amyloid. This perhaps indicates that homocysteine and beta amyloid levels, while both related to cognitive decline, are regulated by independent measures. This RCT did not measure global cognition and memory [66]. Further studies examining homocysteine and amyloid-beta in patients with Alzheimer’s disease are necessary to clarify their relationship to the disease progression.

A 2010 randomized controlled trial found that vitamin B6, B12, and folic acid supplementation decreased the rate of brain atrophy, a characteristic finding in individuals with MCI who later develop Alzheimer’s disease [45].

A 2022 meta-analysis with a total of 95 studies and 46,175 participants found that B vitamins can slow cognitive decline, as measured by score changes in the Mini-Mental State Examination (MMSE) [67]. The interventional period had a significant impact, with B vitamin supplementation greater than 12 months resulting in significant MMSE changes but not so in intervention periods less than 12 months. Additionally, baseline cognitive status had an impact, as only the non-dementia population had slowed cognitive decline from vitamin B supplementation. This last point is contrary to the finding in the previously cited study performed by de Jager et al., which saw a significant benefit of vitamin B treatment in patients with MCI. A separate 2019 meta-analysis of 31 RCTs found no cognitive benefit from the homocysteine lowering effects of B vitamins [68]. Both meta-analyses focused solely on the MMSE as a way to quantitatively measure cognitive function. This ultimately restricted them to examining only this one measurement tool. While encouraging that some studies did find vitamin B supplementation to slow cognitive decline, more trials are needed with a wider range of assessment tools to gain a more comprehensive view of the impact of vitamin B on cognition.

Current research on vitamin B supplementation and cognition has varied in the population sampled, cognition assessment tool, duration of intervention, and type of supplementation, providing only modest evidence to support the use of vitamin B supplementation in cognitive health or dementia. Additionally, there is still some question about the causality between homocysteine and cognitive levels. It is unclear whether increased total homocysteine levels cause cognitive impairment or if high serum homocysteine is a consequence of triggers that result from cognitive decline, such as poor diet and vitamin deficiencies [69]. 

Though the evidence is mixed on vitamin B supplementation and cognitive health, inadequate intake of dietary vitamin B is associated with accelerated cognitive decline [70]. Indeed, one community-based multi-center cohort study found that higher intake of vitamin B, including from dietary sources, correlates with higher cognitive function later in life [71], indicating an important protective role for plant-derived intake of vitamin B.

#### 3.1.3. Vitamin B Mechanism

The benefits of vitamin B on cognitive function is related to the effects on homocysteine. The mechanism by which homocysteine detrimentally impacts brain health is still not fully known. However, it has been hypothesized that increased homocysteine levels result in oxidative stress, increased DNA breakage, decreased methylation of DNA, and dysregulation of its repair [72,73]. These neurotoxic effects are likely what lead to the accumulation of the beta-amyloid proteins and brain tissue atrophy seen in Alzheimer’s disease. Homocysteine can be metabolized via two pathways, either degraded irreversibly or re-methylated to methionine [74]. Homocysteine’s remethylation to S-adenosylmethionine is dependent on vitamins B12, B6 and folic acid [74]. Deficiencies in these B vitamins would prevent the metabolism of homocysteine, resulting in increased levels of homocysteine in the brain.

One consequence of this “homocysteine hypothesis” is that research has directed its focus on B12, folic acid B6, and not as much attention has been given to the other B vitamins. The impacts of the other B vitamins on cognitive function are, as a result, not as well understood. Additionally, it is difficult to determine the extent to which B6 specifically plays a role in cognitive health, as most clinical trials use a combined treatment of B12, folic acid, and B6.

Beyond its role as a necessary cofactor for the metabolism of homocysteine, B6 is also a cofactor in the synthesis of neurotransmitters [44]. Vitamin B6 has also been shown to have an impact on immune function. B6 levels have been inversely associated with systemic markers of inflammation, which is pertinent to note as inflammation contributes to pathologic states like cognitive decline and dementia [75,76]. Ultimately, further research exploring the mechanism of dietary B6 on brain health is necessary to better understand how this plant-derived vitamin can play a role in preventing cognitive decline. 

### 3.2. Antioxidants: Vitamin C, Vitamin E, Beta-Carotene

Oxidative stress is one of the main factors implicated in neurodegenerative conditions like Alzheimer’s disease [77]. Oxidative stress is defined as an imbalance between antioxidants and oxidants, with too much of the latter. The brain is particularly vulnerable to reactive oxygen species, due to its composition of easily oxidizable lipids and high oxygen consumption [77,78]. Mechanistically, reactive oxygen species may augment the production of beta amyloid proteins and the phosphorylation and polymerization of tau proteins, two proteins implicated in Alzheimer’s disease pathology [79,80]. There is current evidence that oxidative stress can be decreased with the consumption of plant-based foods which are high in antioxidants, such as fruit and vegetables [81,82]. Measurements of antioxidant (vitamins C, E, and beta-carotene) levels are higher in individuals on plant-rich diets, perhaps indicating dietary antioxidants as a promising prevention tool for Alzheimer’s disease [83].

#### 3.2.1. Dietary Sources of Antioxidants: Vitamins C, E, Beta-Carotene 

Dietary sources of vitamin C and beta-carotene are from fruits and vegetables, and the main sources of vitamin E are through vegetable oils and nuts [9]. Since humans are unable to synthesize these antioxidants, they are fully obtained through dietary intake.


*Vitamin C*


The best food sources of vitamin C include citrus, kiwi, mango, peppers, tomatoes, and green leafy vegetables [84]. Vitamin C is a water-soluble vitamin, and around 90% of vitamin C daily intake in the general population is from diet, with 5–9 servings of fruit and vegetables estimated to equal 200 mg of vitamin C [84]. Vitamin C is absorbed mostly in the small intestine, through simple diffusion and active transport. In moderate intakes of 30–180 mg/day, vitamin C is absorbed at almost 90% [85].


*Vitamin E*


Vitamin E is found in fat-containing foods, and this fat-soluble property of vitamin E allows it to be stored in fatty tissue so it does not need to be consumed daily. The richest sources of vitamin E are from vegetable oils, although nuts, seeds, and green leafy vegetables also contain high amounts [86]. The benefits of vitamin E are dependent on other vitamins, such as vitamin C. There is a cooperative interaction between these two vitamins, with a combination of vitamin E and vitamin C having a stronger antioxidant effect than either alone [87]. The cooperativity between vitamin E and C may be due to the fact that vitamin C repairs vitamin E radicals, which are formed when vitamin E scavenges oxygen radicals [88].


*Beta-carotene*


Beta-carotene is a fat-soluble vitamin and is the most abundant precursor to vitamin A [89]. Dietary sources of beta-carotene include naturally orange and yellow foods such as carrot, tomato, pumpkin, and papaya [90,91]. The absorption of beta-carotene from plant sources is variable, ranging from 7 to 65% [92]. Dietary fat is one of the major factors that affects beta-carotene absorption, as beta-carotene itself is fat-soluble. A clinical trial showed that uptake of beta-carotene from raw vegetables in salads was significantly increased with the addition of dressings containing higher amounts of fat [93].

#### 3.2.2. Antioxidants Interventions and Current Research

Current studies on supplemented vitamin E, C, and beta-carotene have conflicting results, indicating modest support for the use of supplementation in cognitive health. The Cache County Study, was a cross-sectional and prospective study of 5092 elderly participants, found that a combined use of vitamin E and C was associated with reduced Alzheimer’s disease prevalence [94]. Similarly, the Rotterdam Study, a prospective study of 5393 participants free of dementia, concluded that high dietary intake of vitamin C and E may lower the risk of AD [95]. On the other hand, a prospective study from the Washington Heights-Inwood Columbia Aging Project did not find a decreased risk of Alzheimer’s disease from vitamin C and E intake [96]. These differences in findings may also be due to chance, as the populations in the three studies were similar. The discrepancy points to the need for randomized trials examining the prevention of dementia with antioxidants.

Interestingly, a randomized clinical trial of 78 subjects found that supplementation of vitamin C, E and alpha lipoic acid led to decreased MMSE scores, even with decreased oxidative stress biomarkers in the CSF [97]. An RCT by Lloret et al. had a similar finding, where vitamin E was detrimental to cognition in some patients [98]. However, both these studies had a small sample size (*n* = 78, 57), and this finding has not been confirmed in larger RCTs. A study of 613 patients demonstrated that in patients with mild to moderate Alzheimer’s disease, vitamin E supplementation compared to the placebo resulted in slower cognitive decline [99]. This suggests that vitamin E may have a benefit in slowing disease progression in individuals with Alzheimer’s disease. Additional clinical trials focused on dietary intervention and not supplementation may provide additional evidence of vitamin E’s role in slowing cognitive decline in dementia.

A study by Grodstein et al. found that beta-carotene supplementation had no significant impact on cognition in the short term, but was associated with better verbal memory and overall better global cognitive scores in the long-term [100]. A cross-sectional study found that plasma vitamin C and beta-carotene were significantly lower in individuals with dementia as compared to the control group [101]. While promising that studies have found a positive association between supplemental vitamin C and beta-carotene and cognitive function, the evidence is not strong so more longitudinal studies with larger sample sizes are needed to confirm these effects.

Though the evidence is mixed on antioxidant supplementation and cognitive health, several cross-sectional and cohort studies looking at dietary antioxidant intake and cognition found associations between high food-based intake and better cognitive performance, though other dietary studies did not find similar results [102,103,104,105,106]. However, this may indicate that consuming dietary antioxidants, found in plant-based foods, may support cognitive health.

#### 3.2.3. Antioxidants Mechanism

Antioxidants inhibit cellular damage by donating an electron to reactive oxygen species, effectively neutralizing them and reducing their ability to create damage [107]. The vitamin antioxidants include vitamin E, vitamin C, and beta-carotene [108]. As the body cannot manufacture these antioxidants, it is important to have a diet rich in these nutrients.

Vitamin C is a reducing agent and can neutralize ROS such as hydrogen peroxide, making it neuroprotective against oxidative damage [109]. Imbalance in vitamin C has been linked to neurodegeneration [110]. In addition to its ability to reduce free radicals, vitamin C acts as a first-line antioxidant by promoting regeneration of other antioxidants such as glutathione and vitamin E [111]. The neuroprotective effects of vitamin C are also due to its mitigation of neuroinflammation and suppression of beta-amyloid proteins [112]. In murine models, administration of vitamin C reduced pro-inflammatory cytokines TGF-alpha and IL-1beta as well as ROS [113]. High doses of vitamin C have also been shown to reduce the amount of amyloid plaques in murine models of Alzheimer’s disease [114].

Vitamin E is the major lipid-soluble component of the cell antioxidant defense system [86]. Vitamin E is made up of eight tocopherols and tocotrienols, which are fat-soluble antioxidants. Of these eight, the most highly studied is alpha-tocopherol due to its bioavailability [115]. Vitamin E is located primarily in the cell and organelle membranes and acts as the first line of defense against lipid peroxidation, the process where free radicals degrade the lipid membrane [86,116]. In Alzheimer’s disease, beta amyloid proteins induce oxidative stress which results in protein oxidation and lipid peroxidation, which negatively affects cell signaling and cell membranes [117,118]. Vitamin E can block the production of oxidative species, which decreases the amount of toxicity induced from beta amyloid proteins. Murine models have shown an association between vitamin E deficiency and expression of genes involved in regulation of beta amyloid proteins [119]. Furthermore, tocopherol and tocotrienols have been shown to have an inhibitory effect on enzymes that contribute to neuroinflammation in Alzheimer’s disease [120].

Beta-carotene acts synergistically with other carotenoids in cell and organelle membranes to inhibit lipid peroxidation. One study observed a synergistic cooperativity between beta-carotene and vitamin C in a mechanism similar to that of vitamins B and C, with vitamin C repairing the beta-carotene radical [121]. The benefits of beta-carotene may also be due to its inhibition of acetylcholinesterase (AChE), an enzyme that breaks down the neurotransmitter acetylcholine. Acetylcholine has many functions in the central nervous system, including alertness, learning and memory, and wakefulness [122]. Lower cholinergic function is involved in severity of cognitive dysfunction, and studies have shown that acetylcholinesterase inhibiting drugs treat cognitive symptoms of Alzheimer’s disease [123]. Beta-carotene was found to inhibit AChE in murine models of Alzheimer’s disease, indicating its ability to potentially attenuate cognitive deficits via its antioxidant effects and inhibition of acetylcholinesterase [38].

### 3.3. Vitamin K

Recently, there has been an increased body of evidence that suggests vitamin K has a role in brain physiology [124]. Vitamin K is a fat soluble vitamin that, in addition to its role in blood coagulation, is involved in the metabolism of sphingolipids, a class of lipids involved in the proliferation of brain cells and neuron myelination [124,125]. In addition to its role in brain cell development, vitamin K has been proposed to exert an anti-inflammatory and anti-apoptotic effect in the nervous system [126]. There are two main forms of vitamin K: vitamin K1 (phylloquinone) and vitamin K2 (menaquinone), with the main source of vitamin K1 from green, leafy vegetables and vitamin K2 from animal-based foods, fermented foods and synthesis by gut microbiota. Given that vitamin K2 is from animal sources and that very little is still known about it, we will primarily be examining research on vitamin K1.

#### 3.3.1. Dietary Sources of Vitamin K

Phylloquinone (vitamin K1) is the major dietary source of vitamin K. It is obtained mainly from leafy green plants like spinach and collards. Darker green colored leafy vegetables have higher concentrations of phylloquinone than paler green vegetables, like iceberg lettuce. Green, leafy vegetables contribute approximately 60% of phylloquinone intake [127]. Other plant sources of vitamin K1 include plant oils like soybean, olive, and canola [128].

Menaquinones (vitamin K2) are the product of bacterial fermentation or from the conversation from dietary phylloquinone [129]. Natto, a Japanese soybean dish that is fermented with *bacillus subtilis*, is one of the plant foods highest in vitamin K2 [130]. There is still very little known about the contribution of dietary menaquinones to overall vitamin K levels.

#### 3.3.2. Vitamin K Interventions/Current Research

Studies have shown associations between reduced vitamin K levels and poor cognitive function, however there is still yet to be randomized controlled trials exploring the benefits of vitamin K supplementation on brain health. Multiple small epidemiological studies have examined the relationship between vitamin K, as estimated by food questionnaires, direct measurement of serum vitamin K by high-performance liquid chromatography, and indirect measurements of vitamin K via dephosphorylated uncarboxylated Matrix Gla protein. These epidemiological studies add to evidence that vitamin K may play a promising role in cognitive health [131].

One of the larger cohort studies of 500 participants found that both dietary and serum phylloquinone were strong independent predictors of good cognitive function [132]. These results are in line with murine studies, which have shown vitamin K to have a positive effect on cognition and memory [133,134]. Randomized controlled trials are needed to further explore the relationship between low levels of vitamin K and cognitive decline, but this preliminary evidence supports the potentially protective effect of consuming vitamin K-rich foods on cognitive health.

#### 3.3.3. Vitamin K Mechanism

In recent years, research has shown that vitamin K has an anti-apoptotic and anti-inflammatory effect, specifically mediated by the activation of growth arrest specific gene 6 (Gas-6) and Protein S [135]. Gas-6 is a vitamin K-dependent protein that has a key role in the development of the nervous system and has anti-apoptotic and myelinating activity in neuronal and glial cells [135]. Murine studies have shown Gas-6 to protect hippocampal neurons from apoptosis [136]. Gas-6 has also been found to decrease beta-amyloid induced apoptosis by inhibiting the voltage-gated calcium influx that results in neurotoxicity [137]. Given that beta-amyloid accumulation is a characteristic feature of Alzheimer’s disease, this suggests that vitamin K-dependent Gas-6 may be directly protective for Alzheimer’s disease [137].

Protein S is another vitamin K-dependent protein, and in recent years it has been shown to confer neuronal protection during ischemic injury [138]. Ischemic brain injury has been associated with increased deposition of folding proteins, like the amyloid proteins implicated in Alzheimer’s disease, and some research has even proposed that post-ischemic brain injury may result in Alzheimer’s disease due to the generation of reactive oxygen species [139]. While Protein S does not seem to have the directly protective mechanisms that Gas-6 does, it is likely still beneficial due to its neuroprotective effects.

Currently, clinical studies investigating the role of vitamin K2 and Alzheimer’s disease are lacking. However, in mouse studies, vitamin K2 levels were shown to suppress ROS and decrease the upregulation of proinflammatory cytokines induced by lipopolysaccharides [140]. This suggests that K2 may have some role in reducing neuroinflammation and neurodegeneration. Recent research has focused on the role of the gut microbiome in brain health, with dysbiosis of the gut microbiome linked to poor cognitive health [141,142]. This bidirectional communication between the brain and gut is also known as the “gut–brain-axis”. Dysbiosis has been shown to negatively impact vitamin K production [141]. Given the link between the gut microbiome and Alzheimer’s disease pathogenesis, it is important to further explore the connection between dysbiosis, vitamin K2 and Alzheimer’s disease.

### 3.4. Vitamin D

Vitamin D is a fat-soluble vitamin that plays an essential role in calcium homeostasis and bone growth [143]. In addition to its role in bone growth, vitamin D has vital roles in neurodevelopment [144]. Vitamin D can cross the blood–brain barrier, and calcitriol, the active form of vitamin D, binds to vitamin D receptors (VDR) which are found throughout the brain [145,146]. The presence of high VDR in the human brain during development may indicate vitamin D’s role in neurodevelopment [146]. Vitamin D is primarily synthesized in our skin, in the presence of sunshine. However, for many populations, the main source of vitamin D is through food, such as fatty fish, egg yolks, mushrooms, and foods fortified with vitamin D, and supplements [147]. In this review, we will primarily focus on mushrooms, which are not plants but fungi with a plant-like form [148].

#### 3.4.1. Dietary Sources of Vitamin D

A large number of studies have shown that many countries have suboptimal vitamin D levels, mainly due to lack of sunshine. The main dietary source of vitamin D is from fatty fish, like tuna, mackerel and salmon. Mushrooms that are sun-dried and UV radiation-exposed are also a good source of vitamin D, particularly for vegetarians [149]. While technically a fungus, mushrooms are commonly considered a vegetable in the culinary setting [148]. When exposed to sunlight, the ergosterol that makes up the cell walls of mushrooms is converted into vitamin D [149]. Fresh mushrooms exposed to UV radiation have shown to have high bioavailability of vitamin D, with a 100 g serving of mushrooms providing more than half of daily requirements of vitamin D [150].

#### 3.4.2. Vitamin D Interventions/Current Research 

In a large meta-analysis of 1658 adults without dementia, vitamin D deficiency was shown to be associated with a significant risk of developing dementia [151]. Similarly, in a case–control study, participants with MCI and AD had significantly lower levels of vitamin D compared to healthy participants [152]. These findings are promising as they show that vitamin D may play a role in non-skeletal health.

However, despite these findings, the causal relationship between vitamin D and dementia cannot be confirmed as interventional studies have shown mixed results. A small double-blind placebo-controlled clinical trial showed that a twelve-month supplementation with vitamin D led to improved cognitive function, however a larger double-blind placebo-controlled clinical trial showed that three year supplementation of vitamin D did not improve cognition [153]. Ultimately, while deficient levels of vitamin D are linked to cognitive dysfunction, there is not enough evidence to recommend supplementation of vitamin D to prevent cognitive impairments. More studies, particularly food studies, are needed to examine the relationship between vitamin D and cognitive health.

#### 3.4.3. Vitamin D Mechanism 

The effects of vitamin D are via the binding of vitamin D to an intracellular vitamin D receptor (VDR), which results in the inhibition or transcription of vitamin D-dependent genes [154]. VDR has been shown to be in the brains of humans, rats, mice, and zebrafish [155,156,157]. Given the temporal nature of VDR expression in both mouse and rat brains, it is hypothesized that vitamin D may be important in the differentiation of various cell types in neurodevelopment [144]. Furthermore, rat models have shown that rats born to vitamin D deficient mothers exhibit gross brain morphology and a reduction in nerve growth factor [158].

Vitamin D also may decrease neuroinflammation, due to its antioxidant potential. A rat study showed that vitamin D can increase the levels of glutathione and inhibit inducible nitric oxide synthase (iNOS), both of which reduce the toxicity to neurons [159]. iNOS produces nitric oxide, which is damaging to neurons and oligodendrocytes [160]. By inhibiting iNOS, vitamin D may prevent neuronal damage.

### 3.5. Fiber

Dietary fiber is made up of non-digestible carbohydrates that come from plant foods. Fiber intake has been shown to be associated with lower cholesterol, lower risk of heart disease, enhanced glycemic control, and better gastrointestinal function [161]. There are two main categories of fiber: soluble and insoluble. The main sources of soluble fiber are from fruits and vegetables, and the main sources of insoluble fibers are from whole-grains. Most high-fiber foods have a combination of soluble and insoluble fibers [162]. It is suggested that adults should eat between 20 to 35 g of dietary fiber daily [163].

#### 3.5.1. Fiber and Its Impact on the Gut Microbiome 

Dietary fiber is not broken down by human digestive enzymes but is fermented by gut bacteria, giving rise to short-chain fatty acid (SCFA) metabolites. Acetate, propionate, and butyrate are the primary SCFA products [164]. Recent research has shown SCFAs to have anti-inflammatory effects via modulation of the production of pro-inflammatory cytokines [165,166]. In addition to fiber having immune-modulating effects by production of SCFAs, certain fibers stimulate the immune system directly by interacting with immune cells [167].

Dietary fiber can also influence the composition of bacteria in our gut. Some dietary fiber is classified as prebiotic, which means it is a selective food source for beneficial gut bacteria which stimulates the favorable growth of good gut bacteria, like *bifidobacteria* and *lactobacilli*, while reducing the growth of pathogenic bacteria, like *clostridium* [168,169].

Recently, there has been growing interest in the gut–brain axis. The gut–brain axis is the bidirectional communication between the central and enteric nervous system. Studies have explored the impact of the gut microbiome on cognitive functions. There is emerging showing that the gut microbiota influences levels of anxiety, depression, and autistic behavior [170,171]. Dysbiosis, or an imbalance of the gut microbiome, has also been associated with mood disorders [172]. Studies on germ-free mice have shown how the composition of gut bacteria impacts the expression of neurotransmitters in both the central and enteric nervous system, stress and anxiety, and memory [173,174]. While the importance of dietary fiber on microbiome health has been established, there is still research that needs to be done on the connection between fiber and cognition, likely with the gut–brain axis as a conduit. Additional research on the connection between dietary fiber and cognition may elucidate how plant-based diets rich in fiber may play a role in preventing the progression of Alzheimer’s disease.

#### 3.5.2. Mechanism of Fiber’s Impact on Cognitive Function

While the importance of dietary fiber on gastrointestinal health and metabolism is well established, there is still research that needs to be done examining the impact of fiber on brain processes. The gut–brain axis has emerged as a key communicator between nutrition and the brain. Both microbiota-dependent and microbiota-independent effects of dietary fiber on cognition have been hypothesized.

Independent of the gut microbes, dietary fiber can promote the tight junction protein assembly in the gut, thereby promoting intestinal integrity [175]. A tight gut lining is important as loss of this integrity allows for harmful molecules like lipopolysaccharides (LPS) to enter the bloodstream, which can trigger systemic inflammation and neuroinflammation [176]. Furthermore, widespread inflammation can lead to the breakdown of the blood–brain barrier, which plays a key role in neurodegenerative disorders like Alzheimer’s disease [177].

There are multiple microbiota-dependent pathways by which fiber may influence cognition. One way that fiber may communicate with the brain is by production of SCFAs. SCFAs positively impact the intestinal barrier and modulate the immune system in the gut. Outside the gut, SCFAs may also increase the integrity of the blood–brain barrier [178]. Additionally, prebiotic fiber results in the growth of beneficial gut bacteria like *Bifidobacterium* and *Lactobacillus*, which may influence cognition. Animal studies have shown correlations between certain good gut bacterial species and levels of brain-derived neurotrophic factor (BDNF), a key molecule in memory formation [179]. The production of neurotransmitters can also be influenced by bacterial species like *Lactobacillus* [180]. It is likely that there are multiple microbiota-dependent impacts of dietary fiber on cognition via anti-inflammatory effects of SCFAs and the growth of beneficial bacterial species.

Lastly, dietary fiber may benefit cognition by way of the vagus nerve. The vagus nerve may be activated by certain microbe species and SCFAs [181]. Vagal stimulation is beneficial to cognition as it stimulates BDNF expression and may be associated with improved memory, as shown in one human study [182,183].

#### 3.5.3. Fiber Interventions and Current Research

The importance of a healthy diet and mental health has long been appreciated, with large cohort studies showing an association between a healthy diet and better mental health as well as improved executive functioning [184,185,186]. For example, the Mediterranean diet, which is rich in fiber, has been associated with reduced cognitive decline [187,188]. On the other hand, poor diets with increased intakes of processed foods have been associated with decreased executive functioning [189]. While there is growing knowledge on the impact of diet and cognition, it is unknown how much of these benefits can be associated specifically with dietary fiber.

Currently, most studies on fiber and cognition use animal models. As such, we will mainly be analyzing what we know from murine models. A study by Matt et al. found that both butyrate and dietary soluble fibers were associated with improved neuroinflammation. The study found that mice that were fed a high fiber diet had a changed microbiome and increased production of total SCFA production, particularly butyrate. The mice on the high fiber diet also had decreased expression of pro-inflammatory genes and less inflammatory microglial phenotypes [190]. The results of this study confirmed the results of previous murine studies, where butyrate, a SCFA increased in high-fiber diets, attenuated pro-inflammatory cytokines in microglia [191,192].

Murine models have also focused on the role of fiber in the gut–brain axis. A study by Shi et al. showed that fiber-deprived diets resulted in dysbiosis, which was significantly associated with cognitive deficits, reduced SCFA, and damaged hippocampal proteins [191]. Furthermore, microbiome changes were observed before cognitive impairment in mice with fiber deprived diets, perhaps indicating a causal impact of the gut microbiome on cognitive changes. Another study found that beta-glucan, a soluble fiber found in oats and barley, prevented cognitive impairment induced by a high-fat, fiber-deficient diet (HFFD) [193]. The HFFD resulted in microbiota changes, and even after a short-term beta-glucan supplementation of 7 days, there were microbiota changes before cognitive improvement, similar to the study by Shi et al. 

Human studies have found positive associations between dietary fiber and cognition. One study analyzed data from the US National Health and Nutrition Examination Survey (NHANES) between 2011 and 2014, with a cohort of 1070 older adults, and found dietary fiber positively associated with some components of cognitive function, like word recall, word learning, attention, and language [194]. A smaller cross-sectional study of 65 children showed that dietary fiber was correlated with cognitive performance [194]. Similar results were found in a study with elderly subjects, aged 65 and older [193].

One small randomized control trial of 18 healthy female participants showed moderate increases in cognitive performance and increases of the beneficial microbe *Ruminoclostridium* with a four week supplementation of polydextrose, a dietary fiber [195,196]. This could indicate fiber’s role in modulation of cognition via the gut–brain axis. One clear limitation to this RCT is small sample size. While promising to see positive results in this study, more and larger clinical trials are needed to better interpret the connection between fiber and cognition. Additionally, larger studies with participants exhibiting cognitive decline are needed to investigate the benefits of fiber in dementia, like Alzheimer’s disease.

### 3.6. Discussion

Alzheimer’s disease is the most common form of dementia that is associated with high mortality and morbidity. Several non-modifiable risk factors contribute to the risk of developing Alzheimer’s disease, including older age, genetic polymorphisms, and family history. Multiple non-modifiable risk factors include hypertension, obesity, diabetes, and hypercholesterolemia.

Recently, there has been increased research on the role of dietary and lifestyle factors, such as plant-based diets, and Alzheimer’s disease. Diet seems to play a role in cognition, which suggests that prevention strategies may be possible for Alzheimer’s disease. However, there are still discrepancies between study results, and the lack of long-term clinical trials means definitive conclusions cannot be made. Conflicting results between studies may be due to various factors, such as differences in stage of disease, nutrient measurement techniques, age, and cognition measurement tools, though interestingly one study comparing people consuming animal products to those following vegetarian diets found an increased likelihood for dementia in the meat-eating population, indicating a potentially protective role for plant-based food in the diet [197]. More long-term, large-scale interventions are needed to shed light on the role of plant-derived nutrients and Alzheimer’s disease, to help elucidate the complex pathogenesis of Alzheimer’s disease and to explore how dietary changes can prevent and even treat disease.

The national USDA guidelines for a healthy plate of food recommends at least ½ plant foods (fruits and/or non-starchy vegetables) [198]. Furthermore, multiple national organizations promote plant-based diets. The American Medical Association (AMA), the oldest and largest American physician advocacy group, recently passed a resolution for hospitals to provide plant-based meals and to remove processed meats [199]. This resolution is backed by numerous studies that show the ability of plant-based diets to prevent and even reverse chronic conditions. Both the Alzheimer’s Association and the Physicians Committee for Responsible Medicine recommend vegetables, fruits, and whole grains in the prevention of Alzheimer’s disease [200,201]. More specifically, the National Institute on Aging states that the MIND (Mediterranean—DASH Intervention for Neurodegenerative Delay) diet may reduce risk of Alzheimer’s disease [202]. Table 1 below shows the MIND diet’s plant-based food recommendations and their corresponding nutrients from this review. Finally, the WHO guidelines recommend a Mediterranean diet to reduce the risk of cognitive decline or dementia, as is might help and does not harm, but conclude that vitamins B and E, polyunsaturated fatty acid, and multicomplex supplementation should not be recommended [203].

## 4. Conclusions

Research suggests that a plant-based diet is beneficial for cognitive health and may play a role in the prevention or mitigation of symptoms in Alzheimer’s disease. Currently, it is not possible to establish a causal relationship between vitamin B, antioxidants, vitamin K, fiber, and vitamin D and the development of dementia. Organizations like the American Medical Association recommend plant-based eating habits, and the National Institute on Aging specifically recommends the MIND diet, which is rich in plant foods, for prevention of AD. Adopting a plant-based diet is a low-risk and beneficial lifestyle change to address the maintenance of cognitive health and is potentially a method to help prevent cognitive decline.

## Figures and Tables

**Table 1 medicina-58-01025-t001:** MIND diet recommendations and corresponding plant-based nutrients.

MIND Diet Recommendation	Serving Recommendation	Plant-Derived Nutrients	Brief Summary of Benefits of Plant-Based Nutrients, as Described in This Review
Leafy green vegetables	At least 1 serving/day	Vitamin B9, Vitamin K	Vitamin B9: Metabolism of homocysteine Vitamin K: Anti-inflammatory and anti-apoptotic effects in the nervous system. Vitamin K is involved in the metabolism of lipids involved in the proliferation of brain cells and neuron myelination.
		Vitamin C	Vitamin C: Decreases oxidative stress, which is associated with the beta amyloid and tau proteins implicated in AD. Vitamin C also promotes the generation of other antioxidants which results in decreased neuroinflammation.
		Vitamin E	Vitamin E: Decreases oxidative stress, which is associated with beta amyloid and tau proteins. Vitamin E prevents degradation of the cell membrane and may have an inhibitory effect on the enzymes that result in neuroinflammation.
All other vegetables	At least 2 servings/day	Beta carotene	Beta carotene: Decreases oxidative stress, which is associated with beta amyloid and tau proteins. Beta-carotene is associated with increased acetylcholine levels, a neurotransmitter that is important in learning and memory.
		Vitamin B6	Vitamin B6: Metabolism of homocysteine, which is detrimental to brain health. B6 may also play a role in the synthesis of neurotransmitters and is associated with decreased inflammation.
Berries	At least 2 servings/week	Vitamin C	Vitamin C: Decreases oxidative stress, which is associated with the beta amyloid and tau proteins implicated in AD. Vitamin C also promotes the generation of other antioxidants which results in decreased neuroinflammation.
		Fiber	Fiber: Multiple microbiota-dependent and microbiota-independent mechanisms, including fiber’s modulatory effects on the gut–brain axis, promotion of beneficial gut bacteria, and decreased neuroinflammation by support of the gut lining.
Whole grains	At least 3 servings/week	Fiber	Fiber: Multiple microbiota-dependent and microbiota-independent mechanisms, including fiber’s modulatory effects on the gut–brain axis, promotion of beneficial gut bacteria, and decreased neuroinflammation by support of the gut lining.
		Vitamin B9	Vitamin B9: Metabolism of homocysteine, which is detrimental to brain health.
Beans	3 servings/week	Vitamin B6	Vitamin B6: Metabolism of homocysteine, which is detrimental to brain health. B6 may also play a role in the synthesis of neurotransmitters and is associated with decreased inflammation.
		Fiber	Fiber: Multiple microbiota-dependent and microbiota-independent mechanisms, including fiber’s modulatory effects on the gut–brain axis, promotion of beneficial gut bacteria, and decreased neuroinflammation by support of the gut lining.
Nuts	5 servings/week	Fiber	Fiber: Multiple microbiota-dependent and microbiota-independent mechanisms, including fiber’s modulatory effects on the gut–brain axis, promotion of beneficial gut bacteria, and decreased neuroinflammation by support of the gut lining.
		Vitamin E	Vitamin E: Decreases oxidative stress, which is associated with beta amyloid and tau proteins. Vitamin E prevents degradation of the cell membrane and may have an inhibitory effect on the enzymes that result in neuroinflammation.
Olive oil	—	Vitamin E	Vitamin E: Decreases oxidative stress, which is associated with beta amyloid and tau proteins. Vitamin E prevents degradation of the cell membrane and may have an inhibitory effect on the enzymes that result in neuroinflammation.
		Vitamin K	Vitamin K: Anti-inflammatory and anti-apoptotic effects in the nervous system. Vitamin K is involved in the metabolism of lipids involved in the proliferation of brain cells and neuron myelination.
Mushrooms (not specifically recommended in MIND diet, but included in table to acknowledge vitamin D’s role in cognition)	—	Vitamin D	Vitamin D: Binds to vitamin D receptor (VDR), which plays a role in neurodevelopment and nerve growth factors. Vitamin D may decrease neuroinflammation from increasing glutathione levels and inhibiting iNOS.

## Data Availability

Not applicable.

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
