# Peer review of "Plants, Plants, and More Plants: Plant-Derived Nutrients and Their Protective Roles in Cognitive Function, Alzheimer’s Disease, and Other Dementias"

_medicina, 2022, doi:10.3390/medicina58081025_

Round 1
Reviewer 1 Report
Interesting and complete review. I don't have suggestions or changes to suggest.
Author Response
Thank you for your kind review.
Reviewer 2 Report
Considering the public health impact of Alzheimer’s disease in the developed countries and the lack of effective prevention and treatment intervention, the review by Ding at al. is a welcome synthesis of the current knowledge about the protective role of plant-derived nutrients in cognitive function. The manuscript is well written and the scientific rationale behind the recommending of plant-based nutrition for improving cognition and as a prevention tool for Alzheimer’s is well supported.
Author Response
Thank you so much for your kind review.
Reviewer 3 Report
The review entitled “Plants, plants, and more plants: plant-derived nutrients and their protective roles in cognitive function, Alzheimer’s disease, and other dementias” reported dietary sources, putative mechanisms of neuroprotection of several plant-derived nutrients which could be of interest for preventing AD-related cognitive decline. Albeit there is more doubts than proofs of their in vivo efficacies, it is of high interest for researchers in this field to have an up-to-date summaries of vitamins and fibers’ role in neuroprotection. This manuscript is well written and pleasant to read.
- I suggest to amplify a bit the introduction section, especially deepening the complex etiopathology of AD because it is useful that the readers already know all the pathways involved since the beginning of the review instead of meeting the different pathways at different stages of the manuscript. Furthermore, to empower the ratio of preventing AD and dementias following a proper diet, it is necessary to define the non-disease modifying efficacy of approved drugs and difficulties in finding effective treatments.
- µg and not ug at end of page 5
- please be careful to always report “Alzheimer’s disease” and not truncate to “Alzheimer’s”. Otherwise use always AD.
Author Response
Thank you for thoroughly scrutinizing our manuscript. As requested, we have revised the manuscript and addressed your specific comments in the following ways:
1) The introduction section was modified to include multiple etiopathologies of Alzheimer's Disease including elaboration of the Amyloid, Tau, and oxidative stress/inflammation theories of the disease.
2) A section was added in the introduction addressing treatments for Alzheimer's Disease and the difficulty in finding efficacious treatments to-date, and why it is so difficult to find a cure for AD.
3) Page 5 - ug was changed to μg
4) Alzheimer's Disease was always reported as such, or as "AD", and not truncated to "Alzheimer's"